# Improving musculoskeletal injury surveillance methods in Special Operation Forces: A Delphi consensus study

**Joanne Stannard** [iD]*, **Caroline F. Finch** [iD], **Lauren V. Fortington** [iD]

School of Medical and Health Sciences, Edith Cowan University, Perth, Australia

* j.stannard@ecu.edu.au

## Abstract

Musculoskeletal injury mitigation is a priority in military organisations to protect personnel health and sustain a capable workforce. Despite efforts to prevent injury, inconsistencies exist in the evidence used to support these activities. There are many known limitations in the injury surveillance data reported in previous Special Operation Forces (SOF) research. Such studies often lack accurate, reliable, and complete data to inform and evaluate injury prevention activities. This research aimed to achieve expert consensus on injury surveillance methods in SOF to enhance the quality of data that could be used to inform injury prevention in this population. A Delphi study was conducted with various military injury surveillance stakeholders to seek agreement on improving surveillance methods in SOF. Iterative questionnaires using close and open-ended questions were used to collect views about surveillance methods related to injury case definitions and identifying essential and optional data requirements. Consensus was predefined as 75% group agreement on an item. Sixteen participants completed two rounds of questionnaires required. Consensus was achieved for 17.9% (n = 7) of questions in the first-round and 77.5% (n = 38) of round two questions. Several challenges for surveillance were identified, including recording injury causation, SOF personnel's injury reporting behaviours influencing accurate data collection, and surveillance system infrastructure limitations. Key military injury surveillance stakeholders support the need for improved data collection to enhance the evidence that underpins injury prevention efforts. The consensus process has resulted in preliminary recommendations to support future SOF injury surveillance.

## Introduction

Musculoskeletal injuries impose an extensive burden on military organisations, impacting military capability and having significant financial costs [1,2]. For these reasons, injury mitigation is repeatedly stressed as an organisational and research priority to protect personnel's health and sustain a capable workforce [2,3]. Despite increasing efforts to reduce injuries in the military, little attention has been given to improving the surveillance methods used to collect the necessary data that underpin the scientific foundations of these prevention actions. Our recent

**Data Availability Statement:** All data are in the manuscript and/or supporting information files.

**Funding:** This research was partially funded by the Defence Science Centre Western Australia, grant

number G1004733 (JS). This research was also supported by an Australian Government Research Training Program (RTP) Scholarship (JS). The funders had no role in study design, data collection and analysis, decision to publish, or preparation of the manuscript.

**Competing interests:** The authors declare that they have no competing interests.

systematic review of musculoskeletal injury epidemiology in Special Operation Forces (SOF) highlighted many limitations across studies, such as inaccurate, unreliable and incomplete data collection [4]. The inconsistent surveillance methods used made comparing injury patterns between studies difficult, and results were considered likely to have underestimated the injury burden magnitude. The review's findings [4] are similar to previously raised concerns regarding musculoskeletal injury taxonomy in military injury surveillance [5–10].

Injury surveillance is the continuous and systematic collection, analysis, interpretation, and distribution of injury information [11]. Effective surveillance relies on accurate, reliable and complete data to produce sufficient epidemiology evidence to inform and evaluate preventative action [11]. Surveillance in the military can be a passive or active process. Passive or routine surveillance, such as electronic health systems, can provide gross assessments of the 'big picture' and is useful to direct priorities for further investigations, often by active surveillance whereby injury cases are actively sought. Another challenge in population health surveillance is defining what information is relevant to collect and how. In previous SOF surveillance research, information considered critical to understanding injuries by international surveillance standards [4], such as injury mechanism, was often missing, incomplete or inconsistently recorded. As such, SOF injury surveillance studies often lacked essential and reliable information to inform prevention planning adequately. The aforementioned surveillance issues are not an isolated matter to SOF or the military. Many organisations, such as the World Health Organisation (WHO) and various sporting organisations, have released surveillance guidelines to address these difficulties [12–16].

Presently, there are no published guidelines to support effective injury surveillance in SOF. Quality improvement of surveillance methods and data standards are essential to address the current limitations. There is a clear need for recommendations to improve the scientific foundations and evidence used to inform and evaluate injury prevention activities. A consensus approach to this problem supports a shared decision-making process towards improving surveillance methods. Consensus on injury surveillance methods will encourage a globally consistent and complete approach to collecting injury data and improve knowledge-sharing between nations.

## Aim

This study aims to identify a consensus of opinions between military injury surveillance stakeholders related to data requirements and surveillance methods' relevant to SOF. Based on these consensus opinions, preliminary guidelines for injury surveillance in SOF are presented with a view to their future use.

## Methods

The Department of Defence and Veteran Affairs Human Research Ethics Committee granted ethical approval for this study (approval number: 279–20). All individuals provided informed consent before taking part.

### Study design

This study used a Delphi design, an iterative process whereby sequential questionnaires collect a working group's opinion anonymously and focus on converging the group's collective opinion to achieve a consensus on that topic [17]. A predetermined 75% participant agreement on each item was used to define group consensus [18]. Our Delphi stopping criteria was predetermined as three rounds maximum to avoid respondent fatigue [17].

### Identification of experts

Purposive sampling of experts was used to reach international and domestic stakeholders identified from three fields.

1. Authors of published SOF injury epidemiology research within the last five years (n = 10).

2. Authors of published injury epidemiology research in conventional forces within the last five years (n = 15).

3. Australian Defence Force personnel identified as having organisational experience in public health or preventative medicine within a musculoskeletal field (n = 5).

An invitation email was sent to the potential participants seeking their interest to partake. One reminder email was sent to non-respondents with no further contact thereafter.

### Questionnaire development

An online questionnaire was developed using Qualtrics [19] to ask experts for their views about key injury surveillance aspects related to case definitions, data sources and essential and optional data requirements. To inform the questionnaire, a review comparing surveillance methods in SOF injury epidemiology research was first conducted using the 21 studies from our systematic review [4]. The WHO recommended essential data requirements for injury surveillance were used as a 'gold standard' to compare surveillance information and methods used across studies [11]. The review demonstrated considerable variability across studies. Many WHO recommended essential items were not regularly collected, such as the role of human intent in the injury (reported in 0/21 studies), the place of occurrence (6/21) and injury mechanism (5/21) (S1 Table).

Questions were designed from the WHO surveillance guidelines [11], the International Olympic Committee (IOC) consensus statement on injury surveillance in sport [15] and the current injury classification tool used by the Workplace Health, Safety, Compensation and Reporting database of the Australian Department of Defence [20]. The IOC guidelines relating to injury classification and injury onset were included as it is recognised that military personnel sustain similar injuries to those of sporting populations [21], and therefore such data requirements and methods could also be relevant in a military context. With theoretical information or previous application background, 42 questions were presented to the experts in round one. A mix of Likert scales and ranking responses were used with mandatory responses required. Comment boxes were available for experts to elaborate on responses if desired.

### First-round Delphi questionnaire and process

The first questionnaire predominately focused on determining the importance of specific data requirements in a military and SOF context. The first questionnaire also included questions about experts' qualifications, discipline and experience to provide insight into those contributing to the consensus recommendations. A link to the questionnaire was distributed by email to consenting participants who had 14 days to complete the questionnaire. A reminder email was provided on day 10. Individual responses were grouped to determine if consensus was achieved on an item.

### Second-round Delphi questionnaire and process

The second questionnaire was constructed from the first-round findings. Items that had already reached consensus were removed. Suggested additional data items received in experts'

comments in round one were incorporated. The second questionnaire involved experts rating their agreement to include items (agree or disagree) as essential or optional data requirements, clarifying data item methods and outcome reporting. A group summary of the first-round findings was presented to participants. Likert scale results were presented with frequency distribution percentiles for each data item. Selected comments representative of the group's thematic findings were included in the summary. The second questionnaire and response summary were delivered to experts using the same online platform. Experts were instructed to read and consider the response summary when completing the second-round questionnaire. The results from the second-round were analysed the same as round one. Copies of the full questionnaires are available on request.

## Results

From 30 invitations, 16 experts consented to participate (53.3% response rate). Two invites were a failed email delivery and there was no reason ascertainable for nonparticipation of the remaining 12 invitees. Table 1 presents the consenting expert participants' demographic characteristics. The majority of participants were researchers or clinicians, with a median of 26 years of experience in their respective fields. Expert participants resided across four countries. Approximately 40% of experts had prior experience working with SOF personnel.

### First-round Delphi questionnaire

Sixteen experts completed round one. Table 2 presents the first-round results. Seven of the 39 (17.9%) Likert questions reached consensus related to injury case definitions, age, mode of onset and reporting risk. Experts suggested nine new data items for consideration.

### Second-round Delphi questionnaire

Sixteen experts completed round two (100% participant retention). Table 3 presents the second-round results. Thirty-eight out of 49 (77.5%) questions reached consensus. Items not achieving consensus included three place of occurrence subcategories, two injury intent subcategories, two items related to recording methods for sex and activity causation, two demographic variables suggested as essential data items, and three items suggested as optional data requirements. The Delphi ceased after two rounds for several reasons. Firstly, sufficient information was gained to develop preliminary SOF injury surveillance recommendations with group agreement for approximately 78% of data items. Secondly, it was felt that some data items for which consensus was not achieved, such as developing injury causation and mechanism categorisations, required more extensive discussion beyond the scope of this study. The SOF injury surveillance guidelines informed by the Delphi's outcomes are summarised in Tables 4 and 5.

## Discussion

This study obtained expert opinion on the need to develop SOF injury surveillance guidelines and new knowledge to improve injury surveillance methods in the military and SOF globally. There was strong agreement in support of consistent approaches across nations. Consistent methods between nations will allow the sharing of knowledge and assist in joint efforts towards addressing injury mitigation in military organisations globally. While it is appreciated that a completely standardised approach is not always realistic, with a pragmatic view, we encourage those conducting surveillance investigations to follow the guidelines presented below where possible. These guidelines provide a step forward in supporting accurate, consistent and

**Table 1. The demographic and experience profile of the experts participating in the Delphi rounds (n = 16).**

| Demographical characteristics | |
|---|---|
| Gender | |
| Male | 75.0% (n = 12) |
| Female | 25.0% (n = 4) |
| Country of residence | |
| Australia | 56.7% (n = 9) |
| Belgium | 6.3% (n = 1) |
| The Netherlands | 6.3% (n = 1) |
| United States of America | 31.3% (n = 5) |
| Current role | |
| Clinician | 31.3% (n = 5) |
| Researcher | 50.0% (n = 8) |
| Epidemiologist | 12.5% (n = 2) |
| Public health professional | 6.3% (n = 1) |
| Other | - |
| Education | |
| High school certificate | - |
| Master's degree | 25.0% (n = 4) |
| Doctorate | 68.8% (n = 11) |
| Other | 6.3% (n = 1) |
| Experience years | |
| Median (IQR) | 26 (14–31.5) |
| Eligibility stakeholder group | |
| SOF authors | 18.7% (n = 3) |
| Conventional military authors | 62.5% (n = 10) |
| ADF personnel | 18.7% (n = 3) |
| Military personnel representation | |
| Military or ex-serving | 50.0% (n = 8) |
| Civilian | 50.0% (n = 8) |
| Experience working with SOF personnel | |
| Yes | 37.5% (n = 6) |
| No | 62.5% (n = 10) |

Special Operation Forces (SOF), Interquartile range (IQR), Australian Defence Force (ADF).

reliable injury surveillance in SOF. It is anticipated that these guidelines will require periodic evaluation in response to changes in surveillance needs and capabilities.

## Injury case definitions

Four case definitions were agreed on (Table 3). The case definition choice will provide tiered information for injury frequency as some definitions are more sensitive in recording cases than others [22]. When selecting a case definition, consideration should be given to the investigation's purpose and the data sources from which cases are ascertained. A group preference favoured the 'time loss' definition as it can measure the impact of injuries on capability, a metric of interest to commanders. A group agreement indicated that the applied case definition should be explicitly stated, and the influence of the case definition should be discussed when interpreting results or comparing with studies that have used alternative case definitions.

**Table 2. Experts' responses for in the first-round Delphi questionnaire for SOF injury surveillance data items.**

| LIKERT QUESTIONS | Extremely important | Very important | Moderately important | Slightly important | Not important at all |
|---|---|---|---|---|---|
| **Consistent surveillance** | | | | | |
| 1. How important is it to have consistent methods between nations to conduct injury surveillance in military SOF organisations? | 37.5% (n = 6) | 43.8% (n = 7) | 18.8% (n = 3) | - | - |
| **Injury case definitions** | | | | | |
| 2. How important is it to explicitly report the injury case definition of a surveillance study? | 75.0% (n = 12)* | 25.0% (n = 4) | - | - | - |
| 3. How important is it to consider the reporting behaviours of SOF populations when conducting injury surveillance studies? | 56% (n = 9) | 44% (n = 7) | - | - | - |
| 4. How applicable are each of the above classifications and terminology to categorise injury case definitions to a SOF military context? | **Extremely appropriate** | **Somewhat appropriate** | **Neither appropriate nor inappropriate** | **Somewhat inappropriate** | **Extremely inappropriate** |
| 5. All complaints | 50.0% (n = 8) | 18.8% (n = 3) | 18.8% (n = 3) | 12.5% (n = 2) | - |
| 6. Medical attention | 56.3% (n = 9) | 43.8% (n = 7) | - | - | - |
| 7. Time loss | 87.5% (n = 14)* | 12.5% (n = 2) | - | - | - |
| **Essential data requirements** | | | | | |
| 8. How important is rank as an essential data requirement? | - | 25.0% (n = 4) | 31.3% (n = 5) | 31.3% (n = 5) | 12.5% (n = 2) |
| 9. How important is job/employment codes as an essential data requirement? | 43.8% (7) | 37.5% (n = 8) | 18.8% (n = 3) | - | - |
| 10. How important is years of military experience as an essential data requirement? | 12.5% (n = 2) | 37.5% (n = 6) | 43.8% (n = 7) | 6.3% (n = 1) | - |
| | **Agree** | **Disagree** | **Do not know** | | |
| 11. It is best to record the person's actual age (in whole years) rather than grouping into age brackets. | 93.7% (n = 15)* | 6.3% (n = 1) | - | | |
| | **Male/Female** | **More inclusive** | **Do not know** | | |
| 12. Sex should be recorded as binary data, for example, male or female, or more inclusive, such as intersex or transgender? | 50.0% (n = 8) | 31.3% (n = 5) | 18.8% (3) | | |
| How important is the application of these intent of injury categories? | | | | | |
| 13. Unintentional injury (accidental) | 43.8% (n = 7) | 25.0% (n = 4) | 12.5% (n = 2) | 12.5% (n = 2) | 6.3% (n = 1) |
| 14. Intentional injury (self-harm) | 37.5% (n = 6) | 25.0% (n = 4) | 31.3% (n = 5) | 6.3% (n = 1) | - |
| 15. Assault related injury (deliberate acts of violence against another) | 37.5% (n = 6) | 25.0% (n = 4) | 25.0% (n = 4) | 6.3% (n = 1) | 6.3% (n = 1) |
| How important is the application of these place of occurrence items? | | | | | |
| 16. In garrison | 37.3% (n = 6) | 43.8% (n = 7) | 6.3% (n = 1) | 12.5% (n = 2) | - |
| 17. In combat environments | 43.8% (n = 7) | 31.3% (n = 5) | 12.5% (n = 2) | 12.5% (n = 2) | - |
| 18. Field exercise | 37.3% (n = 6) | 43.8% (n = 7) | 6.3% (n = 1) | 12.5% (n = 2) | - |
| 19. Home | 31.3% (n = 5) | 31.3% (n = 5) | 18.8% (n = 3) | 12.5% (n = 2) | 6.3% (n = 1) |
| 20. Non work-related sites | 25.0% (n = 4) | 31.3% (n = 5) | 18.8% (n = 3) | 18.8% (n = 3) | 6.3% (n = 1) |
| 21. Roads, streets | 25.0% (n = 4) | 31.3% (n = 5) | 18.8% (n = 3) | 18.8% (n = 3) | 6.3% (n = 1) |
| 22. Water bodies or sea | 25.0% (n = 4) | 31.3% (n = 5) | 18.8% (n = 3) | 18.8% (n = 3) | 6.3% (n = 1) |
| 23. How important is it to report the methods used to classify injury type and report the code types included in the data analysis? | 56.3% (n = 9) | 37.5% (n = 6) | 6.3% (n = 1) | - | - |
| | **Extremely appropriate** | **Somewhat appropriate** | **Neither appropriate nor inappropriate** | **Somewhat inappropriate** | **Extremely inappropriate** |
| 24. Is it feasible to use a different injury classification tool to what is routinely used in electronic health systems if it is considered more accurate? | 25.0% (n = 4) | 50.0% (n = 8) | 12.5% (n = 2) | 6.3% (n = 1) | 6.3% (n = 1) |

*(Continued)*

**Table 2.** (Continued)

| LIKERT QUESTIONS | Extremely important | Very important | Moderately important | Slightly important | Not important at all |
|---|---|---|---|---|---|
| 25. How appropriate is the use of the ICD-10-CM to record activity causing injury information in future SOF surveillance studies? | **Extremely appropriate** | **Somewhat appropriate** | **Neither appropriate nor inappropriate** | **Somewhat inappropriate** | **Extremely inappropriate** |
| | 37.5% (n = 6) | 37.5% (n = 6) | 18.8% (n = 3) | 6.3% (n = 1) | - |
| 26. Activity codes should be developed specifically for the military. | 31.3% (n = 5) | 56.3% (n = 9) | 12.5% (n = 2) | - | - |
| 27. Mechanism of injury codes should be developed specifically to suit military and SOF type activities. | 50.0% (n = 8) | 37.5% (n = 6) | 12.5% (n = 2) | - | - |
| **Optional data requirements** | | | | | |
| How important is this application of these individual optional data items? | | | | | |
| 29. Race or ethnicity of the injured person | 12.5% (n = 2) | 6.3% (n = 1) | 50.0% (n = 8) | 12.5% (n = 2) | 18.8% (n = 3) |
| 30. Date of injury | 56.3% (n = 9) | 25.0% (n = 4) | 18.8% (n = 3) | - | - |
| 31. Time of injury | 31.3% (n = 5) | 12.5% (n = 2) | 25.0% (n = 4) | 25.0% (n = 4) | 6.3% (n = 1) |
| 32. Residence of the injured person | - | 6.3% (n = 1) | 6.3% (n = 1) | 56.3% (n = 9) | 31.3% (n = 5) |
| 33. Whether alcohol or illegal substance was a factor | 31.3% (n = 5) | 12.5% (n = 2) | 31.3% (n = 5) | 18.8% (n = 2) | 6.3% (n = 1) |
| 34. The severity of injury (e.g., restricted duty days) | 75.0% (n = 12)* | 18.8% (n = 3) | - | 6.3% (n = 1) | - |
| 35. The disposition of the injured person (e.g., admitted to hospital, discharged) | 56.3% (n = 9) | 18.8% (n = 3) | 25.0% (n = 4) | 6.3% (n = 1) | - |
| 36. How important is it to delineate and record injury events as the first recordable event (initial or index injury) or as a second recordable event (recurrent or subsequent injury)? | 62.5% (n = 10) | 18.8% (n = 3) | 6.3% (n = 1) | 12.5% (n = 2) | - |
| 37. How important do you think it is to record the mode of onset of injury? | 87.5% (n = 14)* | 12.5% (n = 2) | - | - | - |
| 38. How applicable is the below classification and terminology to recording mode of onset in a SOF context? | **Extremely appropriate** | **Somewhat appropriate** | **Neither appropriate nor inappropriate** | **Somewhat inappropriate** | **Extremely inappropriate** |
| a. Acute with a sudden onset, e.g., an ankle fracture sustained from parachuting. b. Repetitive with a sudden onset, e.g., medial tibial stress syndrome from repeated running. c. Repetitive with gradual onset, e.g., degenerative knee osteoarthritis. | 75.0% (n = 12)* | 18.8% (n = 3) | - | 6.3% (n = 1) | - |
| 39. How important is it to assess and report the associated injury risk, e.g., risk ratios or incidence rates? | 75.0% (n = 12)* | 12.5% (n = 2) | 12.5% (n = 2) | - | - |
| 40. How important is it to consider the reporting behaviours of SOF populations when conducting injury surveillance studies? | 56.3% (n = 9) | 43.8% (n = 7) | - | - | - |
| **RANKING RESPONSES** | **Most** | | | | **Least** |
| 41. Rank each case definition you think is most appropriate for recording injuries in SOF populations. | Time loss | | Medical Attention | | All complaints |
| 42. Rank the data collecting methods you feel are most important for injury surveillance in SOF. | Primary collection from a health practitioner | | Patient self-report surveys | | Electronic health system |
| 43. Rank which classification tool is most appropriate to coding typical injuries sustained by SOF personnel | OSIICS V13.1 | | ICD-10-CM | | TOOCS 3.1 |

Special Operation Forces (SOF), Orchard Sports Injury and Illness Classification System, Version 13.1 (OSIICS V13.1), International Classification of Disease, 10th Revision, Clinical Modification (ICD-10-CM), Type of Occurrence Classification System, 3rd Edition Revision 1 (TOOCS 3.1).

* Items achieving the predetermined 75% participant agreement used to define group consensus.

## Essential data item requirements

Eight essential data items were agreed upon for SOF injury surveillance. These items are consistent with the WHO's core data recommendations, with the addition of employment

**Table 3. Experts' responses to the second-round Delphi questionnaire for SOF injury surveillance data items.**

| LIKERT QUESTIONS | Agree | Disagree | |
|---|---|---|---|
| **Consistent surveillance** | | | |
| 1. Passive surveillance should strive to include the agreed essential data items and methods in support of effective surveillance. | 100% (n = 16)* | - | |
| 2. Active surveillance research should strive to follow the agreed methods for essential data items and suited optional data items. | 100% (n = 16)* | - | |
| 3. When passive surveillance is limited and cannot capture all essential data items, it is recommended that active surveillance means prioritise researching the missing information. | 93.7% (n = 15)* | 6.3% (n = 1) | |
| **Injury case definitions** | | | |
| 4. When conducting surveillance research, it is recommended that the case definitions used to categorise recordable cases should be one of<br> a. All complaints- All injuries or physical discomfort are a recordable case, including those not leading to medical attention or restrictions<br> b. Performance impairment†- An injury is a recordable case based on the injury resulting in a negative impact on performance<br> c. Medical attention- An injury is a recordable case based on a soldier seeking medical attention<br> d. Time loss- An injury is a recordable case if the injury results in some form of restricted duty for the injured soldier | **A, B, C** 12.5% (n = 2) | **A, B, C, D** 81.3% (n = 13)* | **Disagree** 6.3% (n = 1) |
| 5. Future research should acknowledge the influence of the applied case definition when interpreting the injury outcomes and when comparing the results of studies with alternative case definitions. | 93.7% (n = 15)* | 6.3% (n = 1) | |
| **Essential data item requirements** | | | |
| 6. Non-binary categories should be offered to record sex. | 56.3% (n = 9) | 43.7% (n = 7) | |
| 7. Injury intent is an essential data item record. | 81.3% (n = 13)* | 18.7% (n = 3) | |
| The following categories of injury intent should be included | | | |
| 8. Unintentional | 93.7% (n = 15)* | 6.3% (n = 1) | |
| 9. Intentional | 68.8% (n = 11) | 31.2% (n = 5) | |
| 10. Assault | 68.8% (n = 11) | 31.2% (n = 5) | |
| 11. Place of occurrence is an essential data item<br>The following categories of the place of occurrence should be included | 100% (n = 16)* | - | |
| 12. In garrison | 100% (n = 16)* | - | |
| 13. In combat environments | 100% (n = 16)* | - | |
| 14. Field exercise | 100% (n = 16)* | - | |
| 15. Home | 81.3% (n = 13)* | 18.7% (n = 3) | |
| 16. Non work-related sites | 87.5% (n = 14)* | 12.5% (n = 2) | |
| 17. Roads, streets | 62.5% (n = 10) | 37.5% (n = 6) | |
| 18. Water bodies or sea | 75.0% (n = 12)* | 25.0% (n = 4) | |
| 19. Urban environments† | 68.8% (n = 11) | 31.2% (n = 5) | |
| 20. None attributed† | 68.8% (n = 11) | 31.2% (n = 5) | |
| 21. Either the ICD or the OSIICS V13.1 should be used to record injury classification. | 81.3% (n = 13)* | 18.7% (n = 3) | |
| 22. Research should report injury classification methods, including the type of injury codes used. | 87.5% (n = 14)* | 12.5% (n = 2) | |

*(Continued)*

**Table 3.** (Continued)

| | | | |
|---|---|---|---|
| 23. The best standard available to record activity causing injury is the 10th revision of the ICD. | 68.8% (n = 11) | 31.2% (n = 5) | |
| 24. The development of standardised activity categories should be prioritised, ensuring a balance between obtaining comprehensive data and efficient data entry | 93.7% (n = 15)* | 6.3% (n = 1) | |
| 25. Research should prioritise the development of standardised injury mechanisms categories, ensuring a balance between obtaining comprehensive data and efficient data entry. | 100% (n = 16)* | - | |
| Should the following demographic variables be considered essential or optional data items to record? | **Essential** | **Optional** | **Not required** |
| 26. Military rank | 37.5% (n = 6) | 56.3% (n = 9) | 6.3% (n = 1) |
| 27. Employment codes | 93.7% (n = 15)* | 6.3% (n = 1) | - |
| 28. Years of military experience | 56.3% (n = 9) | 37.5% (n = 6) | 6.3% (n = 1) |
| **Optional data items** | | | |
| 29. Injury history should be included as part of the optional data set—with this inferring that laterality and date of injury must be collected, and ideally, date of recovery should be collected. | 100% (n = 16)* | - | |
| Should these items be recommended as part of the optional data set? | | | |
| 30. Race or ethnicity of the injured person | 43.8% (n = 7) | 56.2% (n = 9) | |
| 31. Date of injury | 100% (n = 16)* | - | |
| 32. Time of injury | 75%.0 (n = 12)* | 25.0% (n = 4) | |
| 33. Residence of the injured person | 25.0% (n = 4) | 75.0% (n = 4)* | |
| 34. Whether alcohol or illegal substance was a factor | 87.5% (n = 14)* | 12.5% (n = 2) | |
| 35. The severity of injury (e.g., restricted duty days) | 81.3% (n = 13)* | 18.7% (n = 3) | |
| 36. The disposition of the injured person (e.g., admitted to hospital, discharged) | 81.3% (n = 13)* | 18.7% (n = 3) | |
| 37. Equipment being worn at the time of injury† | 93.7% (n = 15)* | 6.3% (n = 1) | |
| 38. Body Mass Index† | 93.7% (n = 15)* | 6.3% (n = 1) | |
| 39. History of number of deployments† | 62.5% (n = 10) | 37.5% (n = 6) | |
| 40. Environmental conditions† | 87.5% (n = 14)* | 12.5% (n = 2) | |
| 41. Terrain† | 93.7% (n = 15)* | 6.3% (n = 1) | |
| 42. Injury recovery time† | 93.7% (n = 15)* | 6.4% (n = 1) | |
| 43. To record injury severity, at a minimum, a measure of time loss (estimated or actual) is recommended. | 100% (n = 16)* | - | |
| **Other considerations to improve surveillance** | | | |
| 44. Open text boxes to collect injury information is recommended to include to assist data collection† | 93.7% (n = 15)* | 6.3% (n = 1) | |
| 45. At a minimum, a summary table should report the number of occurrences for essential data items and any applied optional data item. | 100% (n = 16)* | - | |
| 46. At a minimum, an incidence rate using the exposure denominator 'per working days' or 'per year' should be reported. | 93.7% (n = 15)* | 6.3% (n = 15) | |
| 47. Surveillance studies with a specific focus to investigate injuries associated with military activities should record the exposure amount to the particular activity under investigation. | 93.7% (n = 15)* | 6.3% (n = 1) | |

*(Continued)*

**Table 3.** (Continued)

| | | | |
|---|---|---|---|
| 48. At a minimum, an exposure measurement of time spent conducting the activity is reported to provide a consistent denominator enabling a comparison of risk between military activities. | 100% (n = 16)* | - | |
| 49. It is recommended that investigators acknowledge reporting behaviours as a limitation and impact on conclusions when interpreting or comparing study results. | 100% (n = 16)* | - | |

†Additional data item proposed by expert participants in round-one.

\* Items achieving the predetermined 75% participant agreement used to define group consensus.

International Classification of Disease (ICD), Orchard Sports Injury and Illness Classification System, Version 13.1 (OSIICS V13.1).

information [11]. Employment data provides insight on subgroups most at risk, such as comparing trainees to qualified SOF personnel. This information allows interventions to target specific at-risk subgroups. The previously identified under-researched data items in SOF surveillance research, such as place of injury occurrence and injury mechanism, were agreed as essential data requirements. These data items are essential to identify injury determinants and provide crucial contextual information for prevention action.

There was no consensus on recording sex for epidemiological purposes, such as binary or recommending more inclusive options. Comments received suggested a redundancy in the question because of the homogenous male population in SOF. Contrasting comments referred to the necessity to record sex in alignment with organisational policy, which some countries mandate more inclusive options [23]. At least 18 countries allow transgender personnel to serve openly [24], and SOF roles are increasingly more available for women to enter [25,26]. Thus, more population diversity is likely to eventuate. Both sex and gender identity are increasingly collected in electronic health systems [27] and in the military context [28] to understand and reduce the health discrepancies of minority groups. We recommend that future recording of sex follows organisational policy.

Recommended methods to classify injuries by pathology and anatomy will depend on the surveillance mode. The International Classification of Disease (ICD) is the most common tool in passive surveillance and offers convenience by classifying health information across multiple scopes of practice [29]. One limitation of the ICD is that it is considered less specific than other options to classify musculoskeletal injuries, such as those used in sports surveillance [30,31]. The Delphi results indicated that the Orchard Sports Injury and Illness Classification System, Version 13.1 (OSIICS V13.1), is preferred for classifying common injury types sustained by SOF personnel. The current injury classification tool used by the Workplace Health, Safety, Compensation and Reporting database of the Australian Department of Defence was considered the least preferred [20]. During active surveillance, it is recommended that either the ICD or the OSIICS V13.1 is used. Data can be translated between these tools [31], allowing comparisons between studies using either method; however, this offers less convenience, and detail may be lost when translating data from the OSIICS V13.1 to the ICD. Where possible, the OSIICS V13.1 for active surveillance is preferred where the intent is to seek greater detail on injury diagnosis.

To avoid misclassification bias, we do not recommend collecting injury pathology data unless that information is provided by a suitably qualified clinician able to make a diagnosis. This may occur in a SOF context where personnel have limited access to doctors or physiotherapists. Injury pathology data should also not be collected during self-report surveys, or this data interpreted cautiously, as classification error is likely to occur if personnel are to self-diagnose and data is influenced by recall bias [32,33]. In these instances, we recommend collecting information more broadly to anatomical locations only.

**Table 4. Recommended SOF injury data requirements for passive and active surveillance based on the Delphi consensus.**

| | PASSIVE SURVEILLANCE | ACTIVE SURVEILLANCE | RECOMMENDED SURVEILLANCE METHODS |
|---|---|---|---|
| **Essential data items—the minimum information required for understanding injuries** | | | |
| Age | ✓ | ✓ | Record actual age |
| Sex | ✓ | ✓ | Record in alignment with organisational policy |
| Employment codes | ✓ | ✓ | Record employment categories |
| Injury intent | ✓ | ✓ | Record if the injury is considered unintentional (accidental) |
| Place of occurrence | ✓ | ✓ | The following locations should be offered<br>• Garrison<br>• Combat environments<br>• Field<br>• Home<br>• Non work-related sites<br>• Water bodies, sea |
| Injury classification | ✓ | ✓ | a) The ICD in passive surveillance |
| | ✓ | ✓ | b) The ICD or OSIICS V13.1 is recommended in active surveillance investigations |
| | ✓ | ✓ | c) Report which injury classification system was used to classify injuries |
| | ✓ | ✓ | d) State the type of codes within the system that was used to classify injuries, e.g., the ICD, 10th revision, Clinical Modification, using codes 700–900 |
| Activity causing injuring | ✓ | ✓ | a) Currently, no best standard of practice available |
| | | | b) Concise epidemiological activity categories relevant to SOF populations should be developed |
| Mechanism of injury | ✓ | ✓ | c) Currently, no best standard of practice available |
| | | | d) Concise epidemiological injury mechanisms categories relevant to SOF populations should be developed |
| **Optional data items–information not essential but desirable as considered contextually relevant to military and SOF** | | | |
| Military rank | | ✓ | Record rank |
| Mode of onset | | ✓ | Mode of onset is strongly recommended to record |
| | | ✓ | a) Mode of onset should be recorded as<br>• Acute with a sudden onset, e.g., an ankle fracture sustained from parachuting<br>• Repetitive with a sudden onset, e.g., medial tibial stress syndrome from repeated running<br>• Repetitive with gradual onset, e.g., degenerative knee osteoarthritis |
| Injury history | | ✓ | a) It is strongly recommended to record injury history |
| | | ✓ | b) To do this, record the date of injury, injury laterality and injury recovery time |
| Severity of injury | | ✓ | A measure of time loss (estimated or actual) is recommended, e.g., record days of restricted duty required |
| The disposition of the injured person | | ✓ | Consider the relevance to the study aim, e.g., was the person admitted to a hospital, required outpatient treatment only |
| Time of injury | | ✓ | Record as binary (day/night) |
| Whether alcohol or illegal substance is a factor | | ✓ | a) Consider relevance to record for the study aim |
| Equipment being worn | | ✓ | a) Consider relevance to record for the study aim |
| Environmental conditionals/terrain | | ✓ | a) Consider relevance to record for the study aim |
| Body mass index | | ✓ | Record number |

✓ indicates data item is required for active or passive surveillance modes.

International Classification of Disease (ICD), Orchard Sports Injury and Illness Classification System, Version 13.1 (OSIICS V13.1), Special Operation Forces (SOF).

There was no group agreement on methods to record activity causation. The use of the ICD external cause codes as best practice was contested. An unequivocal group agreement indicated that developing concise activity causation and injury mechanism codes is a priority to

**Table 5. The recommended outcome reporting for all surveillance based upon the outcomes of the Delphi consensus process.**

| Reporting variables | Methods |
| --- | --- |
| Outcome results | All essential data items and the used optional data items should report a summary of the number of occurrences in table format |
| Measurement of risk | a) Passive or active surveillance should report an incident rate |
| | b) Active surveillance investigating specific military activities should report an incident rate associated with an exposure dosage denominator |
| | c) A recommended exposure measurement of time spent conducting the activity is recommended to enable a comparison between activities |
| Underreporting of injury | Consider discussing reporting behaviours as a limitation or the likely influence when interpreting outcome results |

support future surveillance in the military and SOF. Developing setting specific causation codes has been encouraged in other settings to ensure that event or exposure information is adequately recorded [15]. Without this information, prevention opportunities cannot be prioritised as the military activities considered most at risk remain unknown.

## Optional data items

Several additional variables are recommended as useful optional data items. These items are considered contextually valuable but not essential for understanding injuries [11]. The items include rank, injury severity metrics and variables related to internal and external risk factors, such as past injury history, terrain or equipment used at the time of injury. Optional narrative text boxes were also agreed as useful to include during data collection processes. Text boxes can capture contextual information when a system or questionnaire cannot categorise all information.

None of the items relating to the injured person's residence, race and deployment number achieved inclusion agreement. Comments received indicated that race analyses are becoming more common in surveillance to understand health determinants in the military better in some countries. Calls to improve the surveillance of racial health inequalities to improve health discrepancies in public health, sport settings, and health related journals have also been stressed [34–36]. Research exploring injury risk and race in the military is scarce; however, some evidence does exist, suggesting a relationship between race, injuries and injury-related medical discharge [37,38]. Whilst the Delphi results might suggest that recording race is not a priority for SOF injury surveillance, it may also be argued that the absence of evidence means the significance of this information as an injury risk factor is yet to be determined.

## Reporting methods and results

The reporting of surveillance methods should be explicit to improve research validity and comparison of past and future findings. In addition to the Delphi study's findings, we recommend using the Strengthening The Reporting Of Observational Studies In Epidemiology (STROBE) checklist of items to prevent inadequate research reporting [39].

Results for all data items recorded should be reported as a count in table format. Raw data allows for comparison across studies and for future pooled data analyses to occur. Additionally, a risk measurement should be reported indicating the risk associated with an activity. Safe Work Australia, the Australian government body providing oversight on national policy relating to Work Health and Safety, recommends reporting the top three risks associated with the most severe injuries and also for those associated with the most frequently occurring injuries [40]. Risk measurements are essential for comparing activities and informing risk management

prioritisation. Quantifying risk by measuring soldiers' participation in military activities, such as training loads during physical training, is known as exposure data [41]. It is recognised that specific exposure data is not feasible for passive surveillance systems to collect. A more viable approach to quantifying risk for passive surveillance data is to report incidence rates based on a defined number of working days or years as the exposure denominator. Incidence rates provide a simple risk metric for comparison. Active surveillance investigating specific military activities should strive to record exposure dose detailing the exposure time at risk.

## Practical application

Comments received highlighted many challenges within current healthcare systems in conducting surveillance and recommendations to improve surveillance in a broader military context. Surveillance is often limited by software infrastructure and the inability to modify systems to capture desired data. Surveillance systems should be flexible to adapt to evolving data requirements. Modernised systems are required, ensuring optimal data collection is balanced with efficient data entry for users. This is essential for improving surveillance feasibility and acceptability, particularly when clinicians have time restraints and competing demands [42]. It is recommended that surveillance system users undergo training to facilitate correct data entry and understand surveillance rationales to improve data entering compliance [6,42]. Lack of routine distribution of epidemiology information to key stakeholders was also raised as a surveillance challenge. Epidemiology information should be communicated widely to all relevant stakeholders, including commanders or physical training instructors, who are integral to actioning force preservation across the organisation.

Injury reporting behaviours of SOF personnel was a recurring theme discussed as a barrier in collecting accurate information. This is because surveillance often relies upon personnel engaging with military medical systems to collect health information, and SOF personnel are considered less forthcoming to do so when injured. Research in conventional militaries indicates that approximately half of the soldiers within combat units do not seek medical assistance when injured for various reasons, such as fear of negative career consequences [43,44]. Such behaviours will influence data collection and underestimate injury rates. Similar healthcare avoidance behaviours have also been affirmed through emerging SOF research [45]; however, the extent to which SOF personnel do not report their injuries is unknown. These behaviours as a barrier to accurate surveillance is not a novel finding per se and mirrors findings in other occupational and sports settings [46,47], highlighting a common issue that needs to be addressed. These behaviours further justify the importance of investigations where injury cases are actively sought, such as anonymous self-report surveys, to capture injuries not accounted for by passive surveillance. It is recommended that future research investigates the influence of injury reporting behaviours and underreporting frequency in SOF. This information can provide insights into addressing SOF personnel presenteeism, such as targeting engagement or access to healthcare, as well as further insights into the true SOF injury burden size.

The classified nature of SOF operations was infrequently raised as a barrier to data collection. Lack of discussion on this potential limitation could be due to the questionnaire being designed in a non-sensitive manner, and thus when presented to participants, concerns were not raised. Data items where information sensitivity was briefly highlighted as a potential issue related to the place of occurrence and activity causation. One expert's comment suggested that as historical information indicates most injuries occur from physical training, a non-sensitive activity, security requirements are not a significant barrier for effective injury surveillance. Surveillance should always be conducted in alignment with organisational security requirements.

### Limitations and strengths

The strength of this study is the use of a Delphi method as a structured approach whereby experts review and explore a topic, exchange information, and partake in decision making to solve a problem [48]. The anonymity of participants allows individuals to express opinions freely, and the consensus outcome relies on the stability of a group opinion rather than one individual alone. Such research study designs are important for future surveillance methodology research to promote consistency between nations and where informed judgement is required to progress or refine surveillance processes.

Like other Delphi studies, one challenge of this study was recruiting participants [49]. We intentionally contacted 10 authors of previous SOF research as they were ideally placed to provide their opinion. Unfortunately, seven did not participate, with no reason provided. Additionally, although the expert participants are highly skilled in surveillance concepts in a military context, less than half of the participants had experience in SOF. For these reasons, further exploration to develop an agreement on causation codes by a third Delphi round did not occur as this is more suitably completed by experts more familiar with SOF activities. It is recommended that future expert working groups build upon this research to establish military specific activity and mechanism causation data collection methods to address this surveillance gap. This is essential to support future insights into military activities most at injury risk and the subsequent prioritisation of prevention efforts.

There are no statistically bound sample size criteria for Delphi studies. Our sample size is consistent with participant numbers in other Delphi studies [18]. Previous research has indicated that reliable outcomes can be obtained from such sizes when using experts with similar training and knowledge in a given topic with a limited number of experts in a respective field [50]. Given the demographical characteristics of our sample, we believe our sample size was sufficient to achieve reliable results for the purpose of this study.

Experts from 4 different countries participated in our Delphi; subsequently, the expert panel composition was not representative of all nations. Thus, the generalisability of guidelines to other nations may be restricted, such as militaries who do not offer internal healthcare or use electronic health systems. The diverse expert panel, including clinicians, researchers and military members, was a strength of the study, particularly towards finding a feasible solution that compliments real-world practice. An additional strength is that the basis of these guidelines can be adapted to other military populations as similar service populations are likely to need similar surveillance requirements. However, researchers will first need to develop service specific location and causation codes to record crucial contextual information that can direct injury risk management.

## Conclusion

There is a recognised need for improved and standardised data collection to enhance the evidence that underpins injury prevention efforts in SOF. This Delphi consensus process has resulted in preliminary guidelines to improve the data quality in future SOF injury surveillance. Improving methods to record injury causation and mechanism remains a priority for future research. We encourage future surveillance investigations to pragmatically follow the recommendations to support accurate, consistent, and reliable SOF injury surveillance.

## Supporting information

**S1 Table. A summary of the categories of surveillance information recorded by previous SOF injury epidemiology studies according to the World Health Organisation's**

**recommended essential data items.**
(DOCX)

## Acknowledgments

The research team would like to thank the participants who generously shared their time, experience, and knowledge for this research. The opinions expressed herein are those of the author/s and do not necessarily reflect those of the Australian Defence Force/DVA or any extant policy.

## Author Contributions

**Conceptualization:** Joanne Stannard, Lauren V. Fortington.

**Formal analysis:** Joanne Stannard, Caroline F. Finch, Lauren V. Fortington.

**Investigation:** Joanne Stannard.

**Methodology:** Joanne Stannard, Caroline F. Finch, Lauren V. Fortington.

**Project administration:** Joanne Stannard.

**Supervision:** Caroline F. Finch, Lauren V. Fortington.

**Writing – original draft:** Joanne Stannard, Lauren V. Fortington.

**Writing – review & editing:** Joanne Stannard, Caroline F. Finch, Lauren V. Fortington.

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
