## [Decision Letter · Decision Letter 0]

23 Sep 2021

 PGPH-D-21-00367 Improving musculoskeletal injury surveillance methods in Special Operation Forces: A Delphi consensus study PLOS Global Public Health

Dear Dr. Stannard,

Thank you for submitting your manuscript to PLOS Global Public Health. After careful consideration, we feel that it has merit but does not fully meet PLOS Global Public Health’s publication criteria as it currently stands. Therefore, we invite you to submit a revised version of the manuscript that addresses the points raised during the review process.

 Thank you very much for submitting your manuscript to PLOS Global Public Health. This manuscript addresses an important area of research in military health and injury surveillance. Both reviewers highlighted the unique and important contributions that this study makes to the literature. The reviewers recommended some minor revisions, particularly incorporating some additional discussion of the sample size/distribution of respondents, addressing the secrecy of SOF operations as a potential barrier to data collection, further explanation of the impact of the study/of consensus for injury surveillance, and some additional recommendations regarding the translation of these findings into further research. Therefore, I invite you to respond to the reviewers’ comments and revise the manuscript.

In the discussion section, I also would recommend some further elaboration specifically on the global public health implications of these findings.

We look forward to receiving your revised manuscript.

Kind regards,

Kathleen Bachynski, PhD, MPH

Academic Editor

Journal Requirements:

1. Please provide a complete Data Availability Statement in the submission form, ensuring you include all necessary access information or a reason for why you are unable to make your data freely accessible. Note that it is not acceptable for the authors to be the sole named individuals responsible for ensuring data access.

PLOS defines a study's minimal data set as the underlying data used to reach the conclusions drawn in the manuscript and any additional data required to replicate the reported study findings in their entirety. Any potentially identifying patient information must be fully anonymized. 

If your research concerns only data provided within your submission, please write "All data are in the manuscript and/or supporting information files" as your Data Availability Statement."

2. Please amend your detailed Financial Disclosure statement. This is published with the article, therefore should be completed in full sentences and contain the exact wording you wish to be published.

i). Please include all sources of funding (financial or material support) for your study. List the grants (with grant number) or organizations (with url) that supported your study, including funding received from your institution. 

ii). State the initials, alongside each funding source, of each author to receive each grant.

iii). State what role the funders took in the study. If the funders had no role in your study, please state: “The funders had no role in study design, data collection and analysis, decision to publish, or preparation of the manuscript.”

iv). If any authors received a salary from any of your funders, please state which authors and which funders.

Additional Editor Comments (if provided):

Thank you very much for submitting your manuscript to PLOS Global Public Health. This manuscript addresses an important area of research in military health and injury surveillance. Both reviewers highlighted the unique and important contributions that this study makes to the literature. The reviewers recommended some minor revisions, particularly incorporating some additional discussion of the sample size/distribution of respondents, addressing the secrecy of SOF operations as a potential barrier to data collection, further explanation of the impact of the study/of consensus for injury surveillance, and some additional recommendations regarding the translation of these findings into further research. Therefore, I invite you to respond to the reviewers’ comments and revise the manuscript.

In the discussion section, I also would recommend some further elaboration specifically on the global public health implications of these findings.

Reviewers' comments:

Reviewer's Responses to Questions

**Comments to the Author**

1. Does this manuscript meet PLOS Global Public Health’s publication criteria? Is the manuscript technically sound, and do the data support the conclusions? The manuscript must describe methodologically and ethically rigorous research with conclusions that are appropriately drawn based on the data presented.

Reviewer #1: Yes

Reviewer #2: Yes

2. Has the statistical analysis been performed appropriately and rigorously?

Reviewer #1: N/A

Reviewer #2: I don't know

3. Have the authors made all data underlying the findings in their manuscript fully available (please refer to the Data Availability Statement at the start of the manuscript PDF file)?

Reviewer #1: Yes

Reviewer #2: Yes

4. Is the manuscript presented in an intelligible fashion and written in standard English?

Reviewer #1: Yes

Reviewer #2: Yes

5. Review Comments to the Author

Reviewer #1: The discussion of the results demonstrates that the authors really understand the concerns underlying injury surveillance. While I used to work on issues of health in military trainees, I no longer am involved in such work; so the exact findings of the Delphi method are of importance to my day-to-day work. However, the Discussion section puts these recommended inclusions (and those items that the panel did not selection) into such excellent context that this turns into a deep paper on what is needed to undertake injury surveillance. I recommend that this manuscript be accepted.

My only surprise in both the preceding systematic review and the current study is that there was no mention of the secrecy of SOF operations as a potential barrier to data collection. I would defer to the Delphi panel of experts in this case, yet it seems to me that the fact that many SOF missions are Secret or Top Secret (in US military intelligence parlance) that getting exact details of the circumstances of an injury may be somewhat obscured. If the authors do make revisions to the manuscript, I would appreciate some discussion if this has ever come up in this or the prior work.

Also, in Table 3 there is a typo on item 13. It should read n = 16, not n = 100.

Reviewer #2: - Overall / Summary

- This study is a qualitative analysis focused on developing evidence to support a consensus on the reporting of musculoskeletal injury data among military surveillance tools using a Delphi consensus study design.

- The is a very unique study that is well justified and important among the military injury surveillance literature.

- Introduction

- The authors do a great job describing the short comings and inconsistencies that justify the need for the study. However, i think further explanation to what an agreed upon consensus could provide for injury surveillance would be helpful to convey the impact this study may have.

- Methods

- Study design is sound and the authors provided a great description of the methods and steps used

- Is this sample size of interviews considered acceptable among other Delphi studies? it seems somewhat small

- Can the authors please make a justification why they believe this distribution of size of respondents is sufficient for their analysis?

- Results

- clear and concise description of findings from the survey analyses

- All respondents have experience working with military groups but the majority of the respondents do not have a SOF background, however the authors seem to want to make a target consensus within the SOF population. this was addressed sufficiently in the limitations section. however, i think more discussion to the limitations of this study design would be helpful but include how this study design and findings are critical for future studies.

- Discussion

- The authors did a good job discussing the results of the study and how they pertain to practical execution of surveillance techniques. I appreciate the acknowledgment of a smaller agreed upon set of data but allowing optional expansion based on individual needs and ability. However, I think some additional discussion to provide what / how you feel these findings need to be translated to further research will be helpful to convey the progression of this work.

6. PLOS authors have the option to publish the peer review history of their article (what does this mean?). If published, this will include your full peer review and any attached files.

**Do you want your identity to be public for this peer review?** For information about this choice, including consent withdrawal, please see our Privacy Policy.

Reviewer #1: No

Reviewer #2: **Yes: **Nicholas Heebner

---

## [Editor Report · Decision Letter 1]

15 Nov 2021

Improving musculoskeletal injury surveillance methods in Special Operation Forces: A Delphi consensus study

PGPH-D-21-00367R1

Dear Dr. Stannard,

We're pleased to inform you that your manuscript has been judged scientifically suitable for publication and will be formally accepted for publication once it meets all outstanding technical requirements.

Within one week, you'll receive an e-mail detailing the required amendments. When these have been addressed, you'll receive a formal acceptance letter and your manuscript will be scheduled for publication.

An invoice for payment will follow shortly after the formal acceptance. To ensure an efficient process, please log into Editorial Manager at https://www.editorialmanager.com/pgph/ click the 'Update My Information' link at the top of the page, and double check that your user information is up-to-date. If you have any billing related questions, please contact our Author Billing department directly at authorbilling@plos.org.

Kind regards,

Kathleen Bachynski, PhD, MPH

Academic Editor

Additional Editor Comments (optional):

My sincere thanks to both reviewers for their detailed and constructive comments, and to the authors for revising and resubmitting the manuscript. The revisions thoughtfully address the feedback from both reviewers and also include further elaboration on the global public health implications of these findings. I am pleased to recommend acceptance of this article in PLOS Global Public Health.